# A Study of an Algorithm for the Surface Temperature Forecast: From Road Ice Risk to Farmland Application

**Maria Chiara Del Vecchio [1], Alessandro Ceppi [1,\*] , Chiara Corbari [1], Giovanni Ravazzani [1] , Marco Mancini [1], Francesco Spada [2], Enrico Maggioni [2], Alessandro Perotto [2] and Raffaele Salerno [3]**

1   Department of Civil and Environmental Engineering (D.I.C.A.), Politecnico di Milano, Piazza Leonardo da Vinci, 32, 20133 Milano, Italy; mariachiara.delvecchio@gmail.com (M.C.D.V.); chiara.corbari@polimi.it (C.C.); giovanni.ravazzani@polimi.it (G.R.); marco.mancini@polimi.it (M.M.)
2   Ideam, 20090 Cinisello Balsamo (Mi), Italy; francesco.spada@ideamweb.com (F.S.); enrico.maggioni@ideamweb.com (E.M.); alessandro.perotto@ideamweb.com (A.P.)
3   Meteo Expert, 20090 Segrate (Mi), Italy; raffaele.salerno@meteo.expert
\*   Correspondence: alessandro.ceppi@polimi.it

**Abstract:** The presence of road ice has always been a key issue during winter months. A reliable forecast system capable of predicting the Land Surface Temperature (LST) and, consequently, its formation is one of the best strategies to operate towards reducing both vehicles accidents and waste of chemical solvents used for prevention which have a significant economic and environmental impact. Hence, the Meteo Expert Centre (MEC) has developed an algorithm for LST forecasts able to issue ice risk warnings as well. This algorithm operationally works every day in real-time and it is here tested, first, on a paved area of the Pedemontana Lombarda motorway and the Milano Linate airport airstrip, and, afterwards, since the LST plays a crucial role in understanding phenomena of energy exchange between soil, vegetation, and atmosphere, its knowledge and prediction becomes relevant also for other purposes such as agricultural management and irrigation system control, further experiments are carried out over two agricultural fields, one in the North and the other in the South of Italy during the SIM (Smart Irrigation Management) project. All LST analyses showed encouraging results with reasonable high values of statistical scores, in both applications on asphalted and different vegetated terrains, demonstrating that the developed algorithm has a high versatility even on completely different types of surfaces, and it can be applied as a valid tool for road ice risk warnings too.

**Keywords:** LST; road ice risk; temperature forecasts; WRF model; MEC algorithm

## 1. Introduction

The presence of ice on roads has always been a key factor during the cold season: first, it is one of the main causes of road accidents [1–3], then, it has huge environmental and economic impacts due to counteract measures adopted in order to reduce its hazard [4]. Generally, ice is considered more serious than snow on a road, since ice is not often visible, and it does not induce drivers to pay closer attention to road conditions and to moderate their speed. To face and reduce these risks on roads, the main precautionary measure is salt spreading, which is harmful not only for the environment [5], but it also constitutes one of the highest expenses for public administrations both for the cost of chemical solvents and for its highly corrosive properties which, subsequently, require road restorations and interventions.

De-icing operations necessitate strategies that take into account the probability of triggering the phenomenon, considering different modalities how it can occur and including other variables such as

the presence of traffic, sun exposure, and meteorological conditions in the area. Last, but not least, it is important to know the different types of solvents and their mode of action [6], as well as the possible negative effects of their use on soil and water quality [7]. Many types of salt are highly corrosive, and they can both contribute to the deterioration of concrete, steel structures and tires, due to the formation of holes and grains. Dissolution of salt sediments on the roadside may poison the micro fauna and soil in the nearby terrains, which become a hostile environment for plant growth. Its dissolution in water courses may produce changes in the ecosystems and it can be toxic to fishes; in fact, nitrogen and soluble phosphate favor the algae bloom and weed aquatic vegetation.

Nevertheless, salt is commonly used in Italy, where this study is focused on, due to its main action of lowering the freezing point of the water. The presence of salt diminishes the vapor pressure of the solution, which remains in equilibrium with ice at a lower temperature. In particular, as described by the Blagden's law, the freezing-point depression is defined as the difference between the freezing temperature of solvent and the solution (Equation (1))

$$\Delta T_F = K_F \cdot b \cdot i \tag{1}$$

where:

$\Delta T_F$ is the freezing-point depression, defined as TF (pure solvent) − TF (solution).
$K_F$ is the cryoscopic constant, which is dependent on the properties of the solvent, not the solute; for water, $K_F$ is equal to 1.853 °C·kg/mol.
$b$ is the molality (moles solute per kilogram of solvent), in particular 1 mole of NaCl is equal to 58.44 g melt in 1 kg of water.
$i$ is the van't Hoff factor (number of ion particles per individual molecule of solute, e.g., $i$ = 2 for NaCl and 3 for $BaCl_2$).

For instance, to lower the freezing point of water by 2 °C, 2/(1.853 · 2) = 0.54 moles are necessary for each liter of water which equals to 31.45 g of sodium chloride. Let's do a practical example: if we have 10 cm of snow at a temperature of 0 °C which correspond to about 10 mm over a square meter or 10 L/$m^2$, 314.5 g of sodium chloride would occur to lower the freezing temperature by 2 °C. Hence, considering an average cost of 0.07 €/kg for NaCl, the result is a cost of 0.02 € per square meter of surface. This means that for a 6-m width roadway, about € 120 are required for each kilometer. For a city like Milan, where there are about 2000 km of roads, each intervention on the entire network would have a cost of about 240,000 € only for purchasing salt. Hence, a good forecast system of road surface temperature has a noteworthy economic impact and it becomes an important aspect for traffic safety and winter paving maintenance.

This explains why, in recent years, the interest on road ice formation has increased in many countries and networks of sensors have been placed for monitoring the state of the asphalt together with some weather stations for gauging atmospheric conditions. These sensors are capable of measuring the presence of water, ice or snow, surface humidity and temperature, while weather stations measure standard atmospheric variables like air temperature and humidity, wind, and precipitation.

Although the knowledge of real time road conditions plays a relevant interest, the prediction of the same quantities may be crucial for the activation of road winter maintenance service and safety. Therefore, operational weather warning services are needed to mitigate the effects of adverse weather conditions on roads [8]. This requires enough observations and skilful models to predict their conditions.

Since different methods of forecasting the ice risk have been developed, various strategies can be followed focusing the attention on different aspects also depending on the context and on the geographical area. However, two main approaches can be defined for the development of a forecast model: the physically based approach, considering the energy balance equation on the surface for the prediction of the Land Surface Temperature (LST) and the empirical/statistical one. Both the approaches address the same final goal: an accurate prediction of road surface temperature and state

(dry, wet, frost, or ice) which may be a valuable information for authorities and organizations that can decide where and when the interventions have to be put in place [9].

Models based on physical equations of hydro-meteorological processes at road surface have been introduced by several authors in the past [10–13]. For instance, the Deutscher Wetterdienst (DWD) introduced SWIS (Straßenzustands und Wetterinformationssystem) as a nationwide road weather information system, in order to forecast road conditions and surface temperature over a time period of 27 h by an energy balance model [14], while the Finnish Meteorological Institute developed RoadSurf, a modelling system for predicting road weather and surface conditions [15]. Other operational examples exist in many countries as the Model of the Environment and Temperature of Roads (METRo) developed by Environment Canada to predict road conditions and temperatures [16], the road condition model implemented by the Danish Meteorological Institute (DMI) with external forcing by their operational limited area model [17] to produce road weather forecasts for pre-defined sites, which are equipped with observing devices, and the model developed by MeteoConsult in the Netherlands, which has made progress in producing road condition and temperature forecasts for the road network by using the output from an energy balance road model, with high resolution weather forecasts and data concerning sky and sun view factors on the routes [18].

Therefore, it is clear how a reliable system for forecasting the LST and, consequently, the presence of ice is definitely one of the best tools to make the winter road maintenance service effective and efficient, in order to reduce both the frequency of accidents and the waste of salt.

In this paper, we proposed a study of an algorithm developed by the Meteo Expert Centre (MEC) based on land surface temperature (LST) forecasts and, afterwards, applied for predicting ice risk. As described above, several methods have been developed to forecast the LST value, based on the energy balance equations that provide precise results, but involve a large amount of data [19,20]. The MEC algorithm, instead, has the advantage of being fast and operationally easy to apply as it is based on a joint distribution of those variables which influence the road temperature and ice formation. The developed algorithm calculates the expected LST value and an ice alert code between 0 (absence of ice) and 3 (high risk), through a linear function of the weather model output parameters.

This paper is structured as follows: the first part describes the selected areas of the study, the implemented algorithm by MEC, the Weather Research (WRF) meteorological model, and the skill scores used for the assessment of the performance in forecasting the LST and ice risk probability. The second part of the work presents the results obtained for the four test-beds: The Autostrada Pedemontana Lombarda motorway (hereafter abbreviated as "Pedemontana"), the Milano Linate airport (hereafter abbreviated as "Linate"), and the two vegetated areas of the Chiese and Capitanata consortia, while the last section discusses the obtained outcomes about ice warning predictions over the two asphalted surfaces of Pedemontana and Linate during the analyzed winter months.

## 2. Materials and Methods

### 2.1. Area of Study

Four case-studies (Figure 1) are chosen to assess the developed LST algorithm.

The first site area comprises several case-studies analyzed between the years 2017 and 2018 over a track of the Pedemontana motorway in the North-West of Italy at three station points: SP2, Ponte Olona and Gorla (Figure 1a). In this area, observed LST data are measured by Lufft, IRS31Pro-UMB, which provides every 10 min an average temperature of the first centimeters of roadway asphalt with an accuracy of ±0.1 °C for values between −20 °C and +20 °C, else ±0.2 °C. The same sensor can also measure water film height up to 4 mm, freezing temperature for different de-icing materials (NaCl, $MgCl_2$, and $CaCl_2$), road conditions (dry/damp/wet/ice or snow/moist with salt/wet with salt), friction, and ice percentage.

In addition to the road surface sensor, Ponte Olona and Gorla station points are also equipped with a weather station (LUFFT WS600-UMB); hence, the MEC algorithm can be fed with observed

meteorological values which provide the following quantities every 10 min: air temperature and relative humidity, precipitation intensity, type and quantity, air pressure, wind direction, and wind speed.

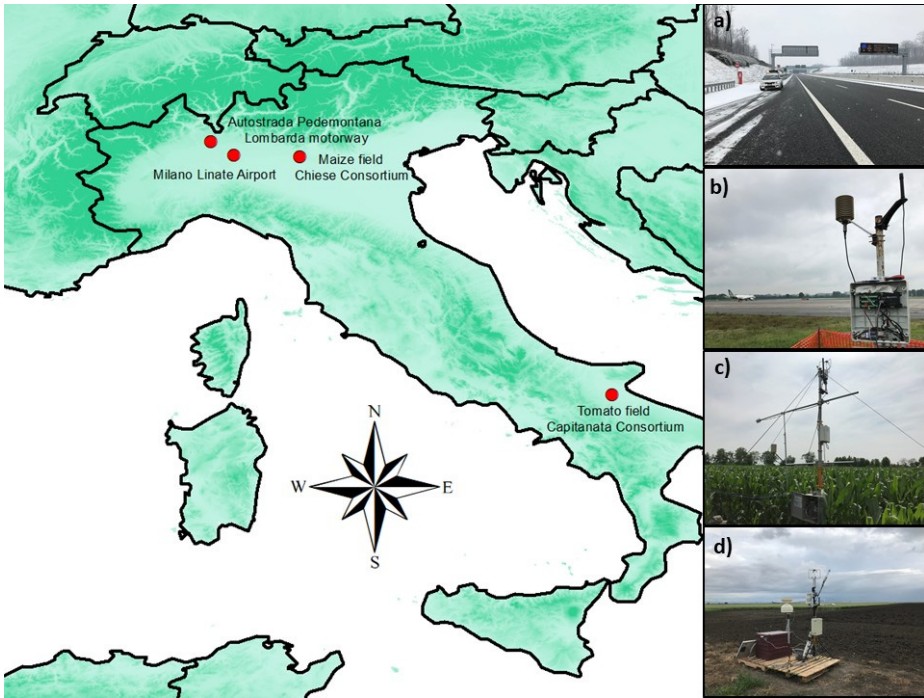

**Figure 1.** The four experimental sites where the Meteo Expert Centre (MEC) algorithm is implemented: The Autostrada Pedemontana Lombarda motorway (**a**), the Milano Linate airport airstrip (**b**), the maize field in the Chiese Consortium (**c**), and the tomato field in the Capitanata Consortium (**d**).

The second site is inside the Milano Linate airport near the runway (Figure 1b) where a dataset of four years (2015–2018) of measurements was available. Observed LST values are semi-hourly measured by contactless sensor sensitive to infrared radiation, the Apogee radiometers SI-111 which has an accuracy of ±0.20 °C for temperatures between −10 °C to 65 °C. In addition, a thermo-hygrometer probe (by Vaisala) is mounted for measuring air temperature and relative humidity, whose values are stored every 30 min as well.

Finally, the LST algorithm performance has also been tested on two non-asphalt surfaces: a maize vegetated area (Figure 1c) in the Chiese river basin (Brescia district, northern Italy), and a tomato field (Figure 1d) in the Capitanata Consortium (Foggia district, southern Italy). In these two latter areas soil monitoring surveys have been undertaken by the research group of the Politecnico di Milano during the growth season of 2016, 2017, and 2018 in the framework of SIM project, Smart Irrigation Management [21], www.sim.polimi.it. Of course, for the two vegetated areas, the analyses have been performed to assess the reliability of the LST prediction in different climate regimes and over cultivated fields, while the prediction of ice risk has been omitted, since data collection are available during growing seasons only. In these last two measurement areas (Chiese and Capitanata consortia), all values are semi-hourly archived trough an eddy-covariance tower, fully equipped both with air and soil instruments (Figure 1c,d); in particular, air temperature, relative humidity, and LST values are monitored with the same sensors used in the Linate airport site.

*2.2. The Weather Model*

The NWP (Numerical Weather Prediction) model used is the WRF-ARW (Weather Research and Forecasting-Advanced Research WRF) version 3.9, developed by the NCAR (National Centre for Atmospheric Research). The model is run operationally by the US National Weather Service and, being open-source and easily portable, it is widely widespread around the world for research and

weather forecasts [22]. The WRF model is designed to be a flexible, state-of-the-art model, and it is developed as a collaborative effort of several institutes. It is supported as a community model and it offers a number of different parameterization for the different physical processes. It is designed to serve both operational forecasting and atmospheric research needs: it features a 3-dimensional and 4-dimensional variational (3DVAR and 4DVAR) data assimilation system, and a software architecture allowing computational parallelism and system extensibility.

The WRF is originally built as a Mesoscale or Local Area Model (LAM), i.e., it simulates the weather in geographical areas smaller than the entire globe. The governing equations, describing the dynamic and thermodynamic evolution of the atmosphere [23], are solved numerically on grid points over the area of interest that constitutes the domain of the model which is three-dimensional. On the horizontal plane models may allow different map projections and utilize a specific grid structure. In the present case, the WRF-ARW employs the Arakawa C-grid [24]. This grid is staggered, which means that not all governing equations are solved on the same grid points: dynamic physical quantities are solved on so-called "flux points", while mass related physical quantities are solved on "mass points". Vertically, a terrain-following coordinate is used [22,23], and vertical levels are denser near the Earth surface and grow more distant from one another when moving towards the top of the atmosphere. The diagonal distance between two mass grid points on the same vertical level defines the model horizontal resolution, which can be chosen by the user too. Local area models require initialization and boundary conditions from a larger LAM or a global model. Initialization data provided the values of all the necessary atmospheric and surface fields at the beginning of the simulation and contour data provided the evolution of these fields outside of the LAM domain.

The set-up of this implemented model can be used in an ensemble configuration with eight WRF instances, each of them with a different physical configuration and with a different set of initial and boundary conditions, which compose the ensemble [25,26]. The ensemble approach has been proved to be a valid instrument in reducing the biases of the physical parameterizations of the model, and in reducing the errors present in the initial and boundary data, especially for meteorological forecasts, which depend on highly non-linear equations [27]. The first-level approach in extracting the information from an ensemble model is the use of average values of the different meteorological parameters: in such a way the meteorological output can be treated as the output of a deterministic forecast. Alternatively, the use of the probabilistic distribution function, which comes as the output of an ensemble forecast system, allows the user to have a full view of all possible scenarios, and to evaluate the probability of occurrence of every single deterministic configuration.

The initial and contour data come from the Global Ensemble Forecast System (GEFS) model with a temporal step of 3-h [28]. Further variability is obtained by using different physical parameterization within the WRF model: different radiation schemes, microphysics schemes, boundary layer schemes, convection schemes, and land surface schemes have been used within the ensemble configuration.

Nevertheless, the ensemble mean is used in this study, since this analysis is related to daily operative applications of LST forecasts by the two companies who manage the Pedemontana motorway and Linate airport, and a single value of temperature forecast has been required. However, the mean of the ensemble scenarios is generally the most accurate forecast [29], and it is more performing than a high resolution deterministic model prediction.

The model has a 6 × 6 km horizontal grid with 37 vertical levels (up to 50 hPa), covering the entire Italian territory with hourly output and a forecast horizon of 5 days. In this paper, the comparison has been carried out on the *day+1* forecast, i.e., the forecast issued today for tomorrow. Finally, the four nearest grid points of the WRF model which encompass the selected site for each case-study are averaged out to obtain a weighted forecast value according to the proposed and described method in [30].

### 2.3. The Implemented Algorithm for LST Simulations

The algorithm developed by MEC aims to predict the Land Surface Temperature (afterwards also called T-skin): it consists of a training algorithm which uses the main quantities connected with ice formation (i.e., air temperature, relative humidity, cloud cover, solar radiation, and wind speed). The algorithm can be divided into two blocks: the first has the forecasted (or observed) air temperature (°C), solar radiation (W/m$^2$), wind speed (m/s), and cloud cover (%) as input, and the estimated LST value (°C) as output; the second receives the output of the first block (forecasted LST) as input, and, combined with the forecasted value of air relative humidity, is able to provide an alert code for ice formation: 0 (no ice), 1 (low risk), 2 (medium risk), or 3 (high risk). Various steps are necessary, because the output of the meteorological model needs to be trained to best represent the diurnal temperature curve and to best consider nocturnal radiative cooling.

According to the algorithm, the evaluation of the ice alert code is performed in four steps, three of which are necessary for the estimation of LST (Equations (2), (3), and (4)), and the last one is used for the computation of the ice alert code. Both the LST values and the ice alert codes are calculated on an hourly basis.

Corrections of the hourly air temperature values, in order to improve the matching and the correlation with the real daily temperature profile. The use of an ensemble model has the consequence of smoothing this daily air temperature profile, an aspect that, especially during the night hours, is crucial to the correct forecast of ice. This correction is operated by a simple algebraic formula, with the correction coefficients obtained from a regression model. An example set of the coefficients are reported in Table 1. The temperature correction is also dependent on cloud cover in Table 2 (the clearer the sky, the greater is the temperature variation between night and day) where there is a set of coefficients related to the cloud cover correction.

$$T_{2m,bis}(t) = T_{2m}(t) - [(coefft1(t) \cdot subpar3(t)] \tag{2}$$

where:

- $T_{2m,bis}(t)$, is the air temperature corrected by the algorithm;
- $T_{2m}(t)$, is the air temperature at time *t obtained from the mode*; and
- coefft1(t) and subpar3(t) are two coefficients obtained from the regression model.

**Table 1.** Coefft1, as a function of the time of the day.

| Time (UTC) | Coefft1 (-) |
| --- | --- |
| 00 | 0.66 |
| 01 | 0.72 |
| 02 | 0.78 |
| 03 | 0.85 |
| 04 | 0.98 |
| 05 | 1.10 |
| 06 | 1.10 |
| 07 | 0.90 |
| 08 | 0.75 |
| 09 | 0.35 |
| 10 | −0.02 |
| 11 | −0.25 |
| 12 | −0.50 |
| 13 | −0.75 |
| 14 | −0.85 |

**Table 1.** *Cont.*

| Time (UTC) | Coefft1 (-) |
|:----------:|:-----------:|
| 15 | −0.80 |
| 16 | −0.70 |
| 17 | −0.55 |
| 18 | −0.25 |
| 19 | −0.05 |
| 20 | 0.25 |
| 21 | 0.50 |
| 22 | 0.55 |
| 23 | 0.60 |

**Table 2.** Subpar3 (t) values.

| Cloud Cover (%) | Subpar3 (-) |
|:---------------:|:-----------:|
| $0 \leq$ cloud cover $< 20\%$ | 1.30 |
| $20\% \leq$ cloud cover $< 40\%$ | 1.10 |
| $40\% \leq$ cloud cover $< 60\%$ | 0.90 |
| $60\% \leq$ cloud cover $< 80\%$ | 0.65 |
| $80\% \leq$ cloud cover $\leq 100\%$ | 0.25 |

Calculation of an intermediate version of surface temperature that takes into account the incoming short-wave solar radiation. The main driver of surface temperature, apart from the air temperature, is radiation forcing: incoming short-wave radiation increases surface temperature during the day, and outgoing long wave radiation decreases surface temperature during the night. To take into account this effect, again a simple algebraic formula has been considered, with the use of a set of parameters, obtained through a regression model. The training of the algorithm with the use of real in-situ observation permits to obtain T-skin values which are more realistic and representative of micro-scale climatological effects than the T-skin values obtained directly from the NWP model.

$$Tsk(t) = T_{2m,bis}(t) = \left( \frac{SW_{down}(t) - \alpha}{\beta} \right) \tag{3}$$

where:

Tsk (t) is the output T-skin, $T_{2m,bis}(t)$ is the air temperature obtained at the previous step, $SW_{down}$ is the incoming short-wave radiation, $\alpha$ and $\beta$ are coefficients obtained from the regression model.

In the particular case of Linate airport and Pedemontana motorway both the coefficients have been set to 50 W/m$^2$, which means an average increase in the skin temperature of 1 °C for every 50 W/m$^2$ of incoming short-wave solar radiation, with an estimated maximum increase value of 20 °C in the case of 1000 W/m$^2$. Having a set of observations of the skin temperature, it is possible to train the algorithm with different types of soil cover, like green, agricultural areas, etc.

Correction of the surface temperature calculated in the previous step, in order to take into account the wind effect; the higher the winds, the more the surface temperature tends to match to the air temperature. In this case, the correction formula is a weighted mean between the air temperature and the surface temperature, and the values of the weights have been obtained comparing the forecasted temperatures in different windy conditions.

$$Tsk_2(t) = a \cdot Tsk(t) + b \cdot T_{2m,bis}(t) \tag{4}$$

where:

$Tsk_2(t)$ is the final version of skin temperature, also called LST, *Tsk (t)* is the surface temperature obtained in the previous step, and *a* and *b* are the coefficients reported in Table 3.

**Table 3.** Values assumed by *a* and *b* coefficients.

| Wind_Speed (m/s) | a (-) | b (-) |
|---|---|---|
| $0 \leq$ wind_speed $< 3$ m/s | 1 | 0 |
| 3 m/s $\leq$ wind_speed $< 5$ m/s | 0.8 | 0.2 |
| 5 m/s $\leq$ wind_speed $< 8$ m/s | 0.6 | 0.4 |
| 8 m/s $\leq$ wind_speed $< 12$ m/s | 0.4 | 0.6 |
| 12 m/s $\leq$ wind_speed $< 16$ m/s | 0.2 | 0.8 |
| 16 m/s $\leq$ wind_speed | 0 | 1 |

Once the LST values ($Tsk_2$) have been computed, the ice alert code is evaluated based on a combination of LST and air relative humidity values (UR%), as shown in Table 4. The ice alert code is a combination of the two main factors involved in the ice formation: temperature and presence of water vapor in the air. The values in Table 4 have been calculated using the observed presence of ice on the road, and doing a correlation with the predicted values of surface temperatures and air moisture. The choice of three different alert levels is based on the categorization of three probability levels (low, medium and high) of ice formation, which is an easy and clear instrument for the notification of the alert to motorway operators.

**Table 4.** Ice Alert code algorithm.

| | $Tsk_2 \leq -6.8\,°C$ | $-6.8\,°C \leq Tsk_2 < -3.1\,°C$ | $-3.1\,°C \leq Tsk_2 < 0\,°C$ | $Tsk_2 \geq 0\,°C$ |
|---|---|---|---|---|
| **UR $\geq$ 95%** | 3 | 2 | 1 | 0 |
| **95% > UR $\geq$ 80%** | 2 | 2 | 1 | 0 |
| **80% > UR $\geq$ 40%** | 2 | 2 | 1 | 0 |
| **40% > UR $\geq$ 5%** | 1 | 1 | 0 | 0 |

*2.4. Field Campaigns*

Since one of the aims of this work is to evaluate the performance of the algorithm for the ice alert code forecast, observed skin temperatures are compared with simulated and forecasted data, calculated by the same algorithm forced with observed and forecasted values, respectively, as input. As above-mentioned, in the areas of the Milano Linate airport, Chiese and Capitanata Consortium, observed LST data are measured by contactless sensors, while in the Pedemontana motorway there are contact instruments which measure an average temperature of the first centimeters of roadway asphalt. Therefore, in the case of Pedemontana, the observed data are corrected in order to obtain a more accurate value of skin temperature. This operation was performed thanks to several field campaigns at different times and weather conditions during which the comparison was carried out between observed data coming from the LUFFT sensors and measured values coming from a thermal camera (FLIR, T450Sc, which has an accuracy of 1 °C or ±1% at limited range and ±2 °C or 2%, whichever is greater, at 25 °C nominal) sensitive to infrared radiation.

From these comparisons it emerged that the skin temperature is lower than the one measured by LUFFT sensors, and the difference varies according to the cloud cover, the day time and the location of the station point. For this reason, the correction of the LST observed by LUFFT sensors has been calculated for time slots and for each station point. For the sake of brevity, it can be summarized that:

- for cloudy days, the correction is approximately about 2 °C ± 0.1 °C;
- for clear-sky days the difference between the temperatures is much more marked in the early morning hours where the correction has an average value of about 3 °C ± 1.5 °C, while in the afternoon is of 1.4 °C ± 0.7 °C.

It can be observed that there is not a unique correction value for each time slot, but a range of values between a min/max. Hence, comparisons described in Section 4 show minimum and maximum values of the corrected LST.

### 2.5. Statistical Indexes

The reliability of the MEC algorithm for the LST prediction has been assessed comparing the observed LST data with both forecasted and simulated values. Different skill scores commonly known in scientific literature [31–33] are here used for the performance evaluation, in particular, the Nash-Sutcliffe efficiency, Root Mean Square Error, and the Pearson and determination coefficients.

The Nash-Sutcliffe Efficiency (NSE) explains the capability of the forecast to predict the observed time series, and its formula is expressed in Equation (5):

$$NSE = 1 - \frac{\sum_1^n \left(x_{i,oss} - \hat{x}_i\right)^2}{\sum_1^n \left(x_{i,oss} - m_{oss}\right)^2} \tag{5}$$

where:

n is the observations number, $\hat{x}_i$ is i-th simulated or forecasted value, and $m_{oss}$ is equal to the average value of observations.

If NSE = 0, then the model forecast is no more accurate than the mean of the observations; if NSE < 0 the mean observed value is a more accurate predictor than the model. The Root Mean Square Error measures the "average" error, weighted according to the square of the error, while the Pearson Correlation Coefficient addresses the question how well did the forecast values correspond to the observed values. This is sensitive to outliers and it does not take forecast bias into account.

Finally, the determination coefficient summarizes in a single value how much the analyzed quantity differs from the regression model: the more the index approaches the unit, the better is the performance of the model used. In addition to the LST prediction, the estimation of the ice alert code probability is valued through a contingency table (Figure 2), in order to evaluate the algorithm with familiar skill scores such as the accuracy, bias, false alarm ratio (FAR), miss alarm ratio (MAR), and hit rate (HR).

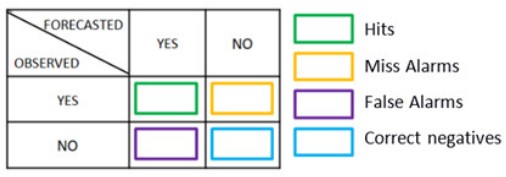

| Index | Range | Best score | Formula |
|---|---|---|---|
| Accuracy | 0 ; 1 | 1 | $Accuracy = \dfrac{Hits + Correct\ Nevatives}{Total}$ |
| BIAS | 0 ; +∞ | 1 | $Bias\ score = \dfrac{Hits + False\ Alarms}{Hits + Miss\ Alarms}$ |
| False Alarm Ratio | 0 ; 1 | 1 | $False\ Alarm\ Ratio = \dfrac{False\ Alarms}{Hits + False\ Alarms}$ |
| Miss Alarm Ratio | 0 ; 1 | 1 | $Miss\ Alarm\ Ratio = \dfrac{Miss\ Alarms}{Hits + Miss\ Alarms}$ |
| Hit Rate | 0 ; 1 | 1 | $Hit\ Rate = \dfrac{Hits}{Hits + Miss\ Alarms}$ |

**Figure 2.** Contingency tables and statistical indices used in the evaluation for the ice alert code probability.

The accuracy, which is simple and intuitive, can be misleading, since it is heavily influenced by the most common category, usually "no event" in the case of infrequent weather patterns. To resolve this issue, only days with air temperatures below an arbitrary threshold of 4 °C were selected in the whole dataset for the performance evaluation of ice warning probability: hence, the number of "no event" has been largely reduced.

As shown in Table 4, the algorithm issues four different ice alert codes between 0 (no ice) and 3 (high risk). To simplify the comparison, and to create a 2 × 2 contingency table, only one level of ice warning is considered, independently from its value.

Finally, to better highlight the differences between different sky conditions, statistical indexes are calculated dividing the dataset in function of the ratio between the observed incoming solar radiation and the radiation we would measure at clear sky conditions: if the ratio remains higher than 0.75, most

of the radiative transfer is able to occur, and we call it "clear day" and if it is lower than 0.25, most of the radiative transfer is not able to occur, and we call it "cloudy day".

Intermediate conditions i.e., partly cloudy days, are here omitted because an accurate forecast of the incoming and outgoing solar radiation is a hard task [34]. This decision has been taken also considering that the majority of ice formation cases in northern Italy happens during clear sky nights, and so the choice to avoid partly cloudy days does not undermine the applicability of the algorithm. Further improvements in the predictability of cloud formation need to be done in order to implement the algorithm in every sky condition.

## 3. Results and Discussion

The obtained results are discussed in two sections: the first detailing the performance of the algorithm which forecasts the LST and the second analyzing the probability of ice risk—the latter for the Pedemontana and Linate case-studies only.

### 3.1. Algorithm Performance for LST Forecasts

This section details with the performance of the MEC algorithm in forecasting LST. This is carried out for all the four cases-study through two steps:

- the validation of the LST algorithm by comparison the observed and simulated LST, using measured data as forcing input into the algorithm.
- the performance evaluation carried out by comparison the observed and forecasted LST using the WRF model as forcing input into the algorithm.

### 3.1.1. The First Case-Study: The Pedemontana Motorway

This section illustrates the Pedemontana motorway case-study. The dataset covers a period of twenty-one months from 1 April, 2017 to 31 December, 2018. As mentioned before, to estimate the algorithm performance, surface temperature data (measured by the LUFFT IRS31Pro-UMB sensor) are compared with simulated data forced with observed quantities (measured by LUFFS WS600-UMB sensor), and with the forecasted ones using the WRF meteorological model as input.

The following Figures 3 and 4 show the scatterplots over the Ponte Olona station point between observed and simulated/forecasted data during clear and cloudy days. For the sake of brevity, similar pictures for the other two station points along the motorway (the SP2 and Gorla) are not shown here, but a complete overview has been summarized in Tables 5–8. The algorithm performance has been tested in two configuration sets: the first one uses hourly data (observed and forecasted) as input, the second introduces a moving average value of the three previous hours as input for the incoming solar radiation only, maintaining the other hourly observed/forecasted input unchanged. This choice prevalently depends on the heat absorption by road pavement which is not instantaneous, but it has a certain inertia on surface temperature which is clearly affected by the heat received in the previous hours.

As mentioned in Section 3, the results are shown both considering the minimum and maximum value of LST. In particular, red dots are representative of comparison between simulated/forecasted LST and the maximum value of corrected observed LST, while blue dots indicate the results obtained by the comparison between simulated/forecasted LST and the minimum value of corrected observed LST.

Whether the simulations are carried out using the observed or forecasted hourly solar radiation or the 3-h moving average as input, a general underestimation of skin temperatures is found during clear sky days. However, if we focus the attention on those values close and below 0 °C (significant for ice risk), an overestimation of the simulated and forecasted LST is seen. On cloudy days, instead, a general overestimation is present in the LST, simulated and forecasted, and both using the hourly solar radiation or the 3-h moving average as input.

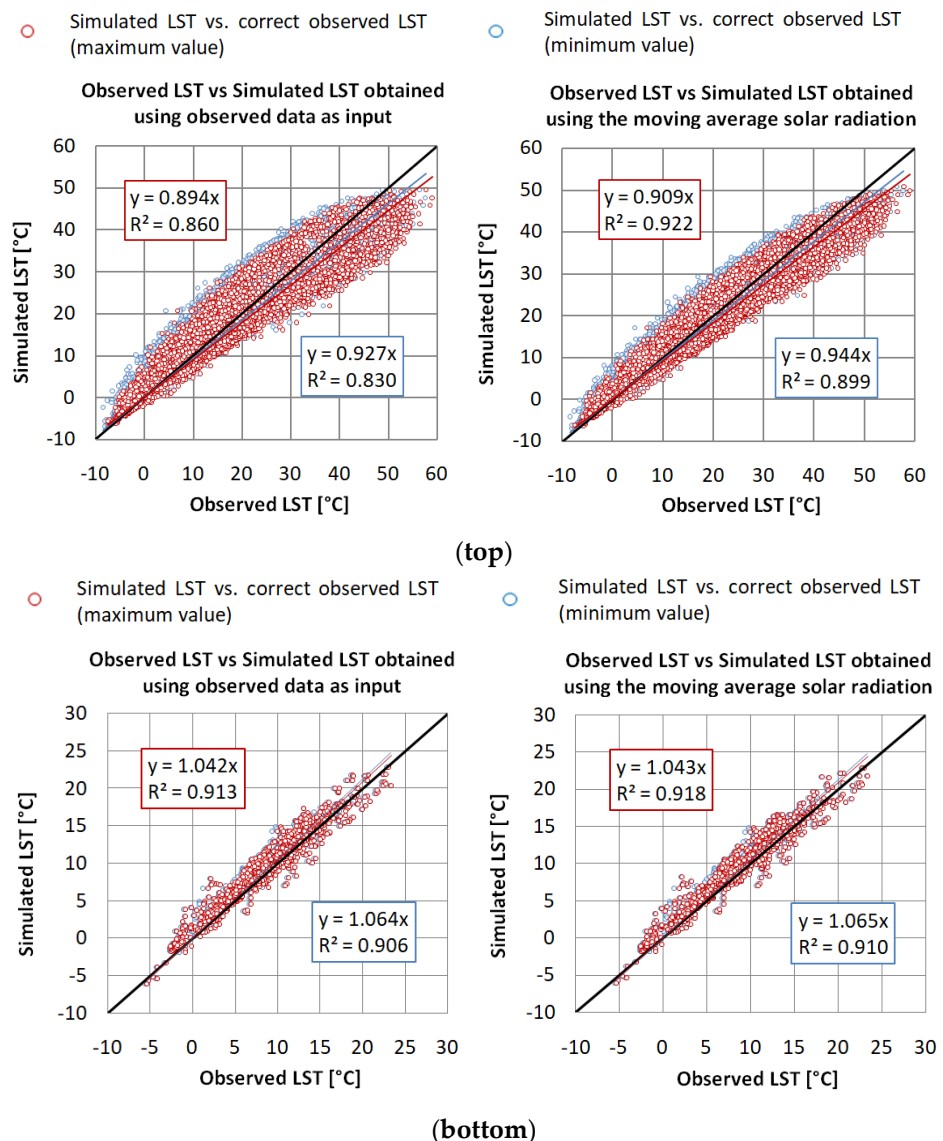

**Figure 3.** Scatterplot between observed vs. simulated Land Surface Temperature (LST) over the "Ponte Olona" station point during clear sky days (**top**). Scatterplot between observed vs. simulated LST over the "Ponte Olona" station point during cloudy days (**bottom**).

The following Tables 5 and 6 indicate the skill scores between observed and simulated data during clear and cloudy days, respectively, while Tables 7 and 8 display the comparison between observed and forecasted data by the WRF model over the analyzed station points. Simulations in the SP2 station point are not available due to lack of a weather station in that track of motorway, hence the algorithm cannot be fed with observed meteorological values, but with forecasted ones only.

Data summarized in Tables 5–8 show a higher performance when the algorithm receives as input the 3-h moving average of solar radiation in the subset of clear sky days, while no sensible differences are found during cloudy days. In particular, the LST simulations compared with observed data put in evidence these score values: NSE index between 0.89 and 0.92, RMSE between 4.3 and 5 °C, Pearson Correlation coefficient between 0.95 and 0.97, and determination coefficient between 0.89 and 0.93 during clear sky days. On the other hand, the NSE index between 0.90 and 0.93, RMSE between 1.4 and 1.7 °C, Pearson Correlation coefficient equal to 0.95, and $R^2$ equal to 0.94 are obtained during cloudy days.

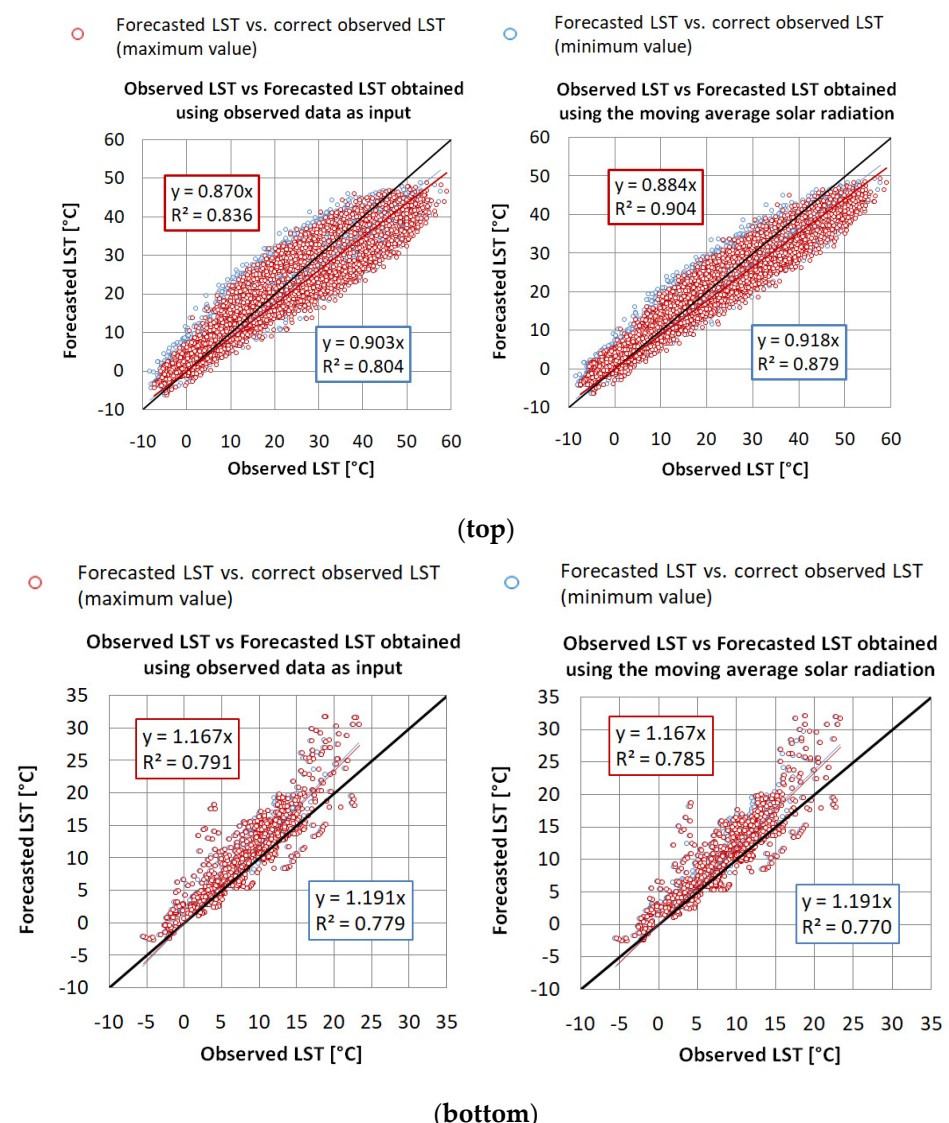

**(top)**

**(bottom)**

**Figure 4.** Scatterplot between observed vs. forecasted LST over the "Ponte Olona" station point during clear sky days (**top**). Scatterplot between observed vs. forecasted LST over the "Ponte Olona" station point during cloudy sky days (**bottom**).

**Table 5.** Statistical indexes during clear sky days between observed and simulated LST values by the MEC algorithm using the hourly solar radiation and the 3 h-moving average as input.

| Index | Station Point | Hourly Solar Radiation as Input | Moving Average (of the Previous 3 h) of Solar Radiation as Input |
|---|---|---|---|
| Nash-Sutcliffe Efficiency (NSE) [–] | Ponte Olona | 0.86–0.87 | 0.91–0.92 |
| | Gorla | 0.84–0.85 | 0.89–0.90 |
| Root Mean Square Error (RMSE) [°C] | Ponte Olona | 5.50–5.60 | 4.30–4.40 |
| | Gorla | 5.90–6.10 | 4.80–5.00 |
| Pearson Correlation coefficient (r) [–] | Ponte Olona | 0.93–0.94 | 0.96–0.97 |
| | Gorla | 0.92–0.93 | 0.95–0.96 |
| Determination coefficient ($R^2$) [–] | Ponte Olona | 0.86–0.88 | 0.92–0.93 |
| | Gorla | 0.84–0.86 | 0.89–0.91 |

**Table 6.** Statistical indexes during cloudy days between observed and simulated LST values by the MEC algorithm using the hourly solar radiation and the 3 h-moving average as input.

| Index | Station Point | Hourly Solar Radiation as Input | Moving Average (of the Previous 3 h) of Solar Radiation as Input |
|---|---|---|---|
| Nash-Sutcliffe Efficiency (NSE) [–] | Ponte Olona | 0.90–0.91 | 0.90–0.92 |
| | Gorla | 0.93 | 0.93 |
| Root Mean Square Error (RMSE) [°C] | Ponte Olona | 1.60–1.70 | 1.50–1.70 |
| | Gorla | 1.40–1.50 | 1.40–1.50 |
| Pearson Correlation coefficient (r) [–] | Ponte Olona | 0.97 | 0.97 |
| | Gorla | 0.97 | 0.97 |
| Determination coefficient ($R^2$) [–] | Ponte Olona | 0.93 | 0.94 |
| | Gorla | 0.93 | 0.94 |

**Table 7.** Statistical indexes during clear sky days between observed and forecasted LST values by the MEC algorithm using the hourly solar radiation and the 3 h-moving average as input.

| Index | Station Point | Hourly Solar Radiation as Input | Moving Average (of the Previous 3 h) of Solar Radiation as Input |
|---|---|---|---|
| Nash-Sutcliffe Efficiency (NSE) [–] | SP2 | 0.86–0.87 | 0.90–0.91 |
| | Ponte Olona | 0.84–0.85 | 0.90 |
| | Gorla | 0.84–0.85 | 0.89–0.90 |
| Root Mean Square Error (RMSE) [°C] | SP2 | 5.20–5.40 | 4.40–4.50 |
| | Ponte Olona | 5.80 | 4.70 |
| | Gorla | 5.90–6.00 | 4.80–4.90 |
| Pearson Correlation coefficient (r) [–] | SP2 | 0.94 | 0.96 |
| | Ponte Olona | 0.93 | 0.96 |
| | Gorla | 0.92–0.93 | 0.95–0.96 |
| Determination coefficient ($R^2$) [–] | SP2 | 0.88 | 0.92–0.93 |
| | Ponte Olona | 0.86–0.87 | 0.92–0.93 |
| | Gorla | 0.85–0.87 | 0.91–0.93 |

**Table 8.** Statistical indexes during cloudy days between observed and forecasted LST values by the MEC algorithm using the hourly solar radiation and the 3 h-moving average as input.

| Index | Station Point | Hourly Solar Radiation as Input | Moving Average (of the Previous 3 h) of Solar Radiation as Input |
|---|---|---|---|
| Nash-Sutcliffe Efficiency (NSE) [–] | SP2 | 0.65–0.69 | 0.64–0.68 |
| | Ponte Olona | 0.58–0.62 | 0.56–0.61 |
| | Gorla | 0.62–0.65 | 0.60–0.63 |
| Root Mean Square Error (RMSE) [°C] | SP2 | 3.00–3.20 | 3.00–3.20 |
| | Ponte Olona | 3.30–3.40 | 3.30–3.50 |
| | Gorla | 3.30–3.40 | 3.40–3.50 |
| Pearson Correlation coefficient (r) [–] | SP2 | 0.90 | 0.89–0.90 |
| | Ponte Olona | 0.90 | 0.89 |
| | Gorla | 0.90 | 0.90 |
| Determination coefficient ($R^2$) [–] | SP2 | 0.80–0.81 | 0.80–0.81 |
| | Ponte Olona | 0.81 | 0.79–0.80 |
| | Gorla | 0.81 | 0.80 |

A lower performance in these analyses is only found for the forecasted LST values vs. observed ones during cloudy days, mainly due to errors in solar radiation predictions by the WRF model: in fact, the NSE index drops from about 0.90 to 0.60 and the determination coefficient from about 0.92 to 0.80, respectively for clear and cloudy sky days. Nevertheless, a general satisfactory reliability of the algorithm using both the simulated and forecasted data as input is achieved.

### 3.1.2. The Second Case-Study: The Linate Airport

The second case-study shows the obtained results at the Milano Linate Airport airstrip. The study is carried out with a dataset of four years from 1 April, 2015 to 31 March, 2019 where the algorithm performance in estimating LST values is analyzed with the same strategy of the previous case-study with the exception of no post-processing corrections for the observed dataset, since the Infra-Red Temperature Sensor (IRTS) directly measures the LST. Regarding the solar radiation input, the 3-h moving average values slightly improves the performance, as in the first case-study, hence, the comparison with hourly data are here neglected. The following Tables 9 and 10 summarize the obtained results.

**Table 9.** Statistical indexes during clear and cloudy sky days between observed and simulated LST values by the MEC algorithm.

| Index | Clear Days | Cloudy Days |
|---|---|---|
| **Nash-Sutcliffe Efficiency (NSE) [–]** | 0.90 | 0.82 |
| **Root Mean Square Error (RMSE) [°C]** | 4.36 | 1.97 |
| **Pearson Correlation coefficient (r) [–]** | 0.97 | 0.95 |
| **Determination coefficient ($R^2$) [–]** | 0.94 | 0.90 |

**Table 10.** Statistical indexes during clear and cloudy sky days between observed and forecasted LST values by the MEC algorithm.

| Index | Clear Days | Cloudy Days |
|---|---|---|
| **Nash-Sutcliffe Efficiency (NSE) [–]** | 0.91 | 0.71 |
| **Root Mean Square Error (RMSE) [°C]** | 4.20 | 2.60 |
| **Pearson Correlation coefficient (r) [–]** | 0.97 | 0.91 |
| **Determination coefficient ($R^2$) [–]** | 0.93 | 0.82 |

For the Linate airport case-study, there is a good performance both for simulated and forecasted data with good skill scores especially during clear sky days with the NSE index between 0.90 and 0.91 and a determination coefficient between 0.93 and 0.94. However, even in this case study a slight lower performance is found in forecasting LST values during cloudy days with the NSE equal to 0.71 and $R^2$ to 0.82.

Figures 5 and 6 show the scatterplots for observed vs simulated data, forced with measured observations during clear and cloudy sky days, and for observed vs forecasted values, forced with WRF forecasts, respectively.

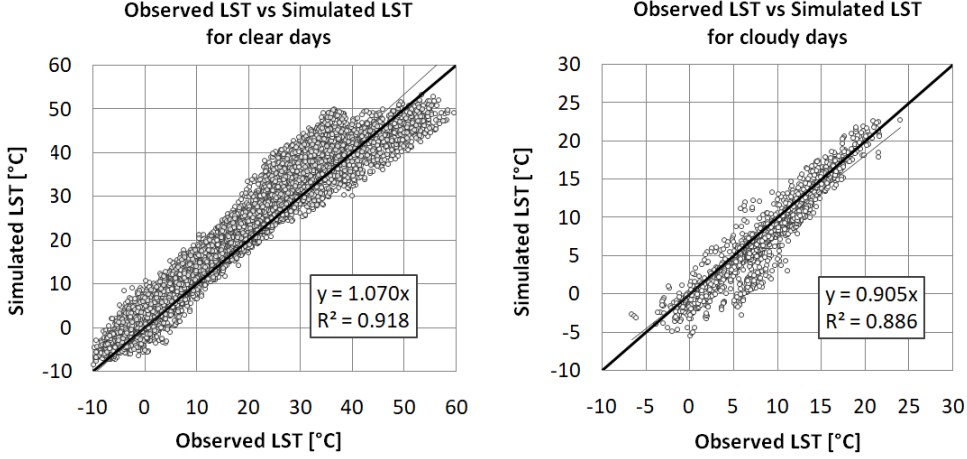

**Figure 5.** Scatterplot between observed vs. simulated LST over the "Linate" station point during clear (left) and cloudy (right) sky days.

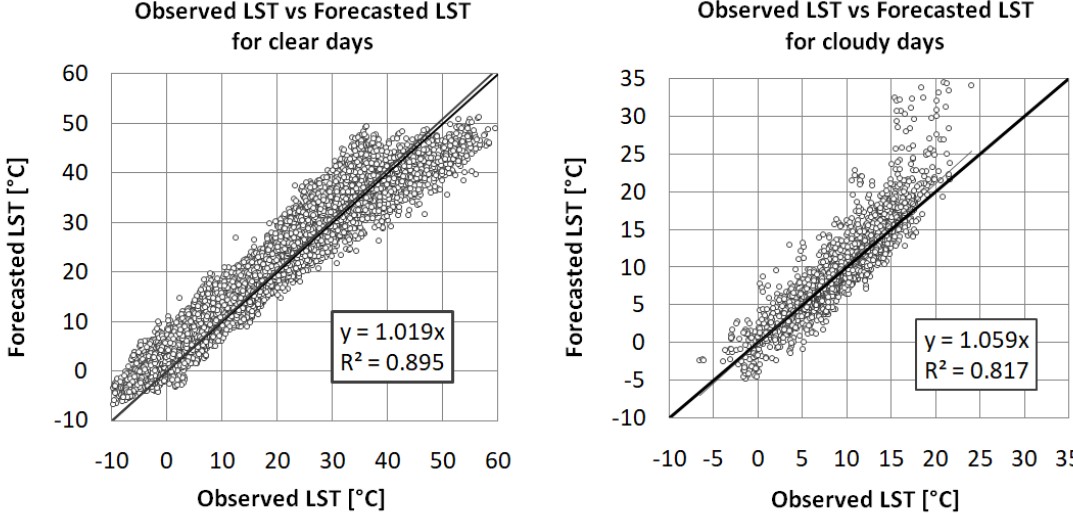

**Figure 6.** Scatterplot between observed vs. forecasted LST over the "Linate" station point during clear (left) and cloudy (right) sky days.

As for the Pedemontana case-study, a general overestimation can be found during clear sky days for simulated and forecasted LST values (Figure 5) when temperatures are close and below 0 °C. Conversely, a slight tendency to underestimate is generally present during cloudy days, even with the lowest temperature values (Figure 6).

### 3.1.3. The Chiese and Capitanata Consortia Test-Beds

The fairly good performance found in the LST simulations and forecasts pushes us to carry on further investigations over different types of surfaces and climate regimes using the same algorithm. Hence, the algorithm performance has been verified over two surfaces different from asphalt: a maize and a tomato field of the Chiese and Capitanata consortia, in northern and southern Italy, respectively, where the Politecnico di Milano has carried out soil monitoring surveys during the SIM project. To test the sensitivity of the MEC algorithm even in cultivated fields, where energy fluxes play a relevant role between earth surface and atmosphere, and the LST prediction may affect the management of water resources, we decided to extend the analysis during growing seasons. Using the same equations described in Section 3, the algorithm reliability leads to poor results as highlighted in Table 11 (red colors). However, by calibrating Equation (4), a remarkable improvement is achieved (green color, in Table 11). In particular, the change of the β coefficient corresponds to a change in the amount of energy required to raise the surface temperature by 1 °C over one square meter of vegetated soil. After an appropriate tuning, we set the β parameter at a constant value of 250 W/m$^2$.

**Table 11.** Statistical indexes obtained before and after the calibration of the β coefficient.

| Statistical Index before/after the Algorithm Calibration | Chiese Consortium | | Capitanata Consortium | |
|---|---|---|---|---|
| | Observations vs. Simulation | Observations vs. Forecasts | Observations vs. Simulation | Observations vs. Forecasts |
| Nash-Sutcliffe (NSE) [–] | −1.24/0.82 | −0.87/0.82 | −1.22/0.84 | −1.07/0.76 |
| Root Mean Square Error (RMSE) [°C] | 7.68/2.43 | 6.94/3.14 | 7.97/2.06 | 7.70/2.63 |
| Pearson Correlation Coefficient (r) [–] | 0.86/0.95 | 0.88/0.87 | 0.92/0.98 | 0.93/0.97 |
| Determination Coefficient (R$^2$) [–] | 0.74/0.91 | 0.77/0.76 | 0.84/0.96 | 0.86/0.94 |

The Figures 7 and 8 highlight the LST trend for observed data in comparison with simulated and forecasted values before and after the calibration of the β coefficient over the Chiese (Figure 7) and Capitanata (Figure 8) consortia, respectively. In both the cultivated fields, the applied tuning sensibly improved the performance and both simulations and forecasts match more closely the measured

data at the two station points. For the sake of brevity, this comparison has been carried out only during clear sky days in the growing season, when the incoming solar radiation is higher; hence, ten non-consecutive days are here selected during two different growing seasons: 2018 for the Chiese and 2017 for the Capitanata, whose average duration is generally about five–six months from April till September both for maize and tomato growing season.

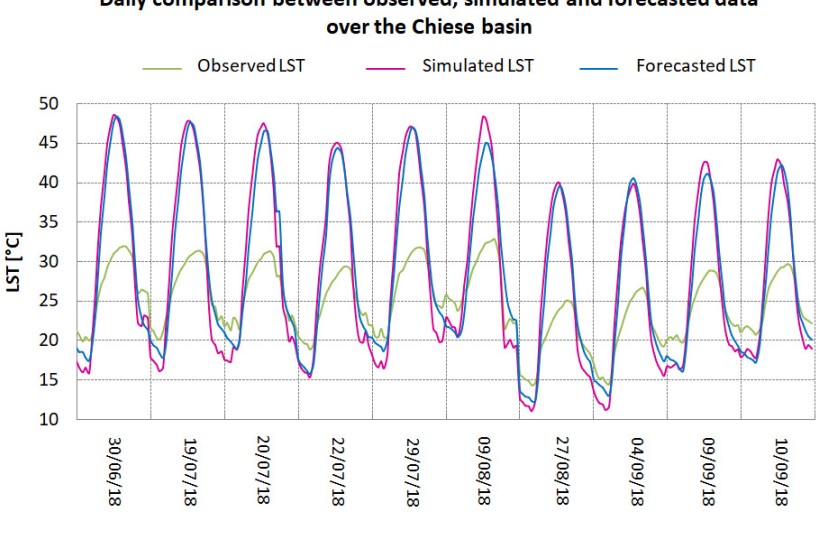

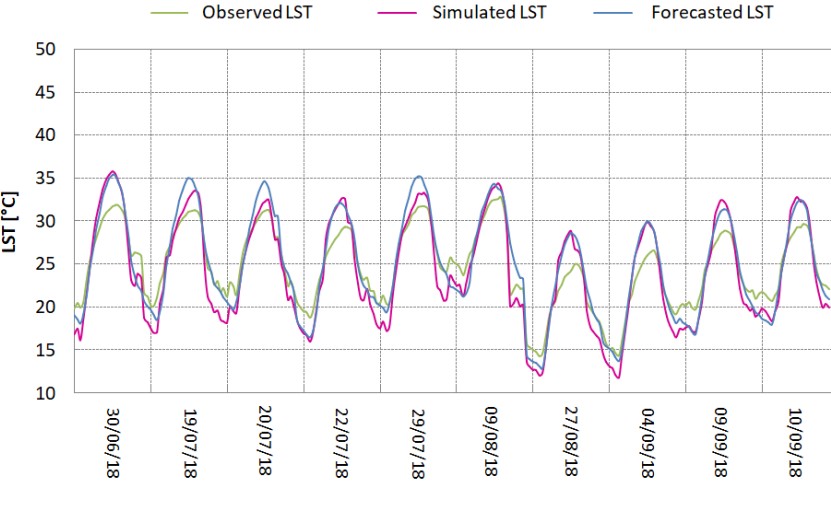

**Figure 7.** Comparison between observed, simulated and forecasted data over the Chiese basin for ten non-consecutive days during the 2018 growing season before (**top**) and after (**bottom**) the calibration.

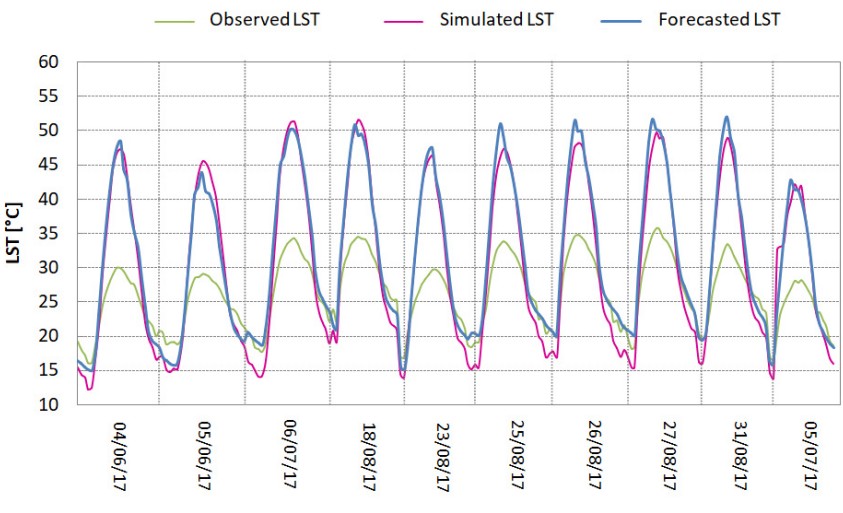

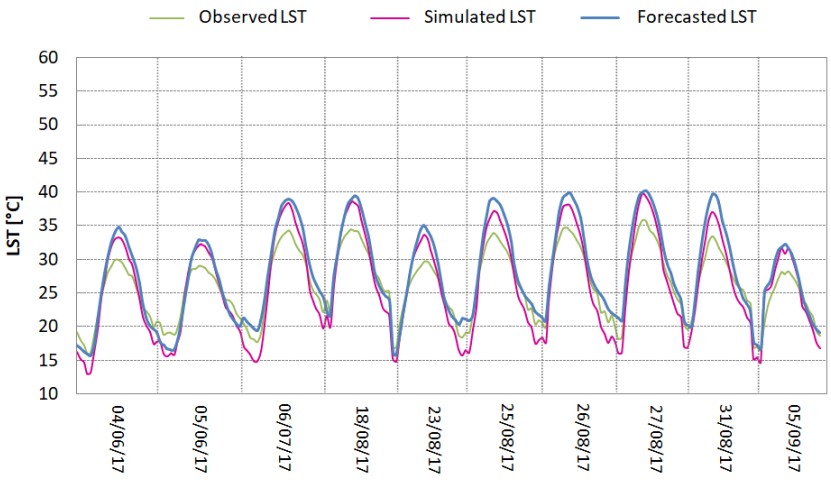

**Figure 8.** Comparison between observed, simulated and forecasted data over the Capitanata consortium for ten non-consecutive days during the 2017 growing season before (**top**) and after (**bottom**) the calibration.

Indeed, these high skill scores found after tuning the MEC-algorithm, give the proof of a high versatility of the implemented model even over different surfaces.

*3.2. Algorithm Performance for Ice-Risk Forecasts*

The second goal of this study tests the performance of the ice risk probability issued by the MEC algorithm. The following sections properly want to highlight the MEC algorithm performance over these two asphalted areas, the Pedemontana motorway and Linate airport, in order to show how accurate LST forecasts are a valuable tool for ice-risk management.

3.2.1. The Case-Study at the Pedemontana Motorway

For the first case-study, the analysis is carried out by comparing the observed road-conditions data, measured by the LUFFT sensors, with the ice alert code forecasted values using the WRF model as forcing input into the algorithm. As mentioned before, since the dataset covers twenty-one months from April 2017 to December 2018 including warmer seasons where no ice risk is expected, only days with hourly air temperature below 4 °C are considered to calculate the contingency tables.

Figures 9 and 10 present the contingency tables during clear and cloud sky days, respectively, for the three station points: SP2, Gorla, and Ponte Olona. It is worth mentioning that the total number of ice risk events during cloudy days is very low, which is consistent with the typical meteorological conditions of this region in northern Italy: in fact, the presence and risk of ice generally is high during clear nights only. This explains equal results in the three neighboring station points under overcast sky conditions.

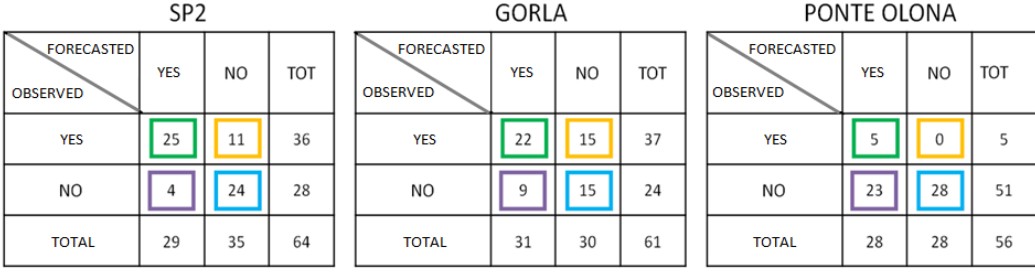

**Figure 9.** Contingency tables for clear sky days.

**Figure 10.** Contingency tables for cloudy days.

All the skill scores are afterwards summarized in Table 12.

**Table 12.** Evaluation of algorithm performance for the ice alert code forecast for Pedemontana motorway: values assumed by the indices.

| Classification | Station Point | Accuracy | BIAS | False Alarm Ratio | Miss Alarm Ratio | Hit Rates |
|---|---|---|---|---|---|---|
| **Cloudy days** | SP2, Ponte Olona, Gorla | 0.93 | 0.80 | 0.00 | 0.20 | 0.80 |
| **Clear days** | SP2 | 0.77 | 0.81 | 0.14 | 0.31 | 0.69 |
| | Ponte Olona | 0.59 | 5.60 | 0.82 | 0.00 | 1.00 |
| | Gorla | 0.61 | 0.84 | 0.29 | 0.41 | 0.59 |
| **Total** | SP2 | 0.81 | 0.80 | 0.12 | 0.29 | 0.72 |
| | Ponte Olona | 0.66 | 3.20 | 0.72 | 0.10 | 0.85 |
| | Gorla | 0.67 | 0.83 | 0.26 | 0.38 | 0.65 |

It clearly appears how the false alarm forecasts at Ponte Olona are quite high. This is mainly due to peculiar microclimate conditions where the weather station is located. In fact, the Ponte Olona station is situated in a track of the Pedemontana motorway on a bridge of the Olona river creek which is subjected to a canyon effect with stronger wind conditions than the other two weather stations located alongside the lane. Notwithstanding this, the obtained results for the other two measurement points (SP2 and Gorla) underline a satisfactory performance, especially during cloudy days as well as highlighted in the LST forecasts; the BIAS index between 0.80 and 0.84 indicates a slight tendency of the MEC algorithm to underestimate the presence of ice.

### 3.2.2. The Case-Study at the Linate Airport

For the ice alert code forecast, in this second case-study, the algorithm performance is studied by comparing the output initially to the joint distribution of the forecasted air relative humidity and LST and, then, both the observed values. As illustrated in Figure 11, 2 × 2 contingency tables are created both for clear and cloudy days and the main statistical indexes are calculated for performance evaluation (Table 13), taking into account only those days where hourly air temperatures are less than 4 °C.

**Figure 11.** Contingency tables for clear (**left**) and cloudy sky days (**right**) for forecasted values.

**Table 13.** Evaluation of algorithm performance in the ice alert code forecast for Linate Airport: forecasted values assumed by the indices.

| Classification | Accuracy | BIAS | False Alarm Ratio | Miss Alarm Ratio | Hit Rate |
|---|---|---|---|---|---|
| **Clear days** | 0.69 | 0.66 | 0.01 | 0.38 | 0.62 |
| **Cloudy days** | 0.86 | 1.00 | 0.20 | 0.20 | 0.80 |
| **Total** | 0.69 | 0.66 | 0.04 | 0.37 | 0.63 |

The analysis confirms a better accuracy for the ice warning probability forecast when it is applied to cloudy days, as for the Pedemontana case-study. On the contrary, during clear skies, it performs slightly worse with a miss alarm ratio equal to 0.38 and a BIAS score less than 1, however, very few cases of false alarm (1%) are observed.

This outcome confirms the good performance found in the previous case-study that leads to the final considerations about the differences in the performance in function of the sky cover. By completely distinguishing clear and overcast sky from those days with variable meteorological conditions, an evident variation in the algorithm reliability between cloudy and clear sky days appears. This has two principal explanations: first, the ice risk during cloudy days is less frequent than the risk on clear nights in this geographical area, and few cases have been analyzed; second, the complex relation between the variables involved in the ice formation produces a decrease in the model performance. In particular, the temperature decrease due to radiation loss is very sensitive to the overall meteorological conditions and to the precise location considered. A small difference in wind speed, e.g., can cause a sensible difference in the mixing of the air in the first few centimeters above the ground, resulting in the formation or in the absence of ice.

## 4. Conclusions

During the period from late autumn to early spring, vast areas of the northern hemisphere experience frequent snow, sleet, ice, and frost. Such adverse weather conditions lead to dangerous driving conditions with consequential effects on road transportation in these areas. A numerical forecasting system is developed for the automatic prediction of slippery road conditions in northern Italy. The system is based on an algorithm forced with meteorological input coming from an operational ensemble prediction system.

This study derives from the need by the Pedemontana motorway and Linate airport to have a tool capable of predicting ice formation on paved surfaces and consequently increasing security and reducing the environmental and economic impact due to the preventive spreading of salt. In particular, one of the aim of this work is the evaluation of an algorithm for ice risk prediction developed by the MEC.

The algorithm is structured in two parts: the first uses as input the results of the WRF model in order to provide as output the forecasted LST values, the second receives as input the LST and the forecasted relative humidity to issue an ice alert code as output. The LST and ice probability forecast performance has been evaluated computing common skill scores.

This analysis has been carried out over four different areas: two asphalted ones, the Pedemontana motorway and the Milano Linate airport airstrip in northern Italy, in addition to two cultivated fields, one in the North of Italy and the other in the South. Results highlight a better performance when the algorithm is applied to asphalt surface. However, despite the fact that the procedure has been designed for paved areas, the experiment on two cultivated terrains, the maize field of the Chiese basin and the tomato fields of the Capitanata Consortium, shows how the system can be easily adapted in different conditions after a suitable tuning of equation coefficients.

The forecast of the ice warnings generally shows good results with a higher reliability especially during cloudy days, due to a smaller number of events in comparison with clear nights. In the latter case, the accuracy tends to diminish and the values of missed alarms are greater than false alarms. Therefore, since the cost of failing to take anti-icing measures when ice does form on the road is much greater than that of anti-icing when no ice forms [35], this outcome requires a deeper investigation in future developments, improving the WRF model parameterization of the heat loss by radiative transfer at the surface, and improving cloud cover forecast.

**Author Contributions:** Conceptualization, A.C., G.R., M.C.D.V., C.C., E.M., and R.S.; software, F.S.; validation, A.P.; formal analysis, M.C.D.V.; investigation, M.C.D.V., A.C., and E.M.; resources, A.P. and E.M.; data curation, C.C. and G.R.; writing—original draft preparation, M.C.D.V.; writing—review and editing, A.C., M.C.D.V., E.M., A.P., R.S., G.R., and C.C.; supervision, M.M. and R.S.; All authors have read and agreed to the published version of the manuscript.

**Funding:** This research received no external funding.

**Acknowledgments:** This study is carried out thanks to the support of the Autostrada Pedemontana Lombarda S.p.A., in particular, Luca Ferraris and Igor Restelli, who allowed us to work in safe conditions during the field surveys, and the Milano Linate airport staff by the name of Andrea Tulipano who gave us a precious support at the airport airstrip. For the same reason, a particular thanks goes to Eng. Luca Cerri and Eng. Gabriele Lombardi who participated and collaborated to the measurement campaigns.

This work was awarded at the third edition of the "Sergio Borghi price 2019", dedicated to all students of Italian universities who carried out their bachelor or master degree in meteorological applications.

Finally, monitoring activities over Chiese and Capitanata consortia mentioned in this paper are part of two projects: SIM (Smart Irrigation Management) www.sim.polimi.it and RET-SIF (Real Time Soil Moisture Forecast for Smart Irrigation) www.retsif.polimi.it.

**Conflicts of Interest:** The authors declare no conflict of interest.

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
