# Peer review of "A Study of an Algorithm for the Surface Temperature Forecast: From Road Ice Risk to Farmland Application"

_applsci, doi:10.3390/app10144952_

Round 1
Reviewer 1 Report
The article is devoted to an actual problem of scientific and especially practical significance. The proposed system of automatic numerical prediction of the Land Surface Temperature (LST) and the probability of ice formation based on the adaptation of the WRF model and instrumental measurements of the parameters of the atmosphere and the earth's surface deserves high praise.
Apparently, the algorithm and forecast system were originally developed for the forecast of LST and ice on roads and runway. But a careful assessment of the forecast accuracy on four different objects (paved areas, highways, runways, cultivated fields) located in different regions of Italy showed the probability of adaptation and applicability of the algorithm for a wider range of practical tasks. Based on statistical analysis, a high reliability of the forecast of ice is established, especially on cloudy days.
With pleasure noting these achievements of the authors, I would like to note the following problems. From the presented text, it is not clear how in the forecast of LST and ice takes into account:
– the influence of precipitation (rain, snow) in the process of their fall;
– the influence of warm and cold fogs;
– the influence of non-uniformity of wind and humidity in air flows.
It is known that mountain-valley and breeze circulation leads to the fact that along the highway can form bands of ice and frost, which differ sharply in the conditions of ice formation on the surfaces.
This indicates the complexity and multiplicity of the problem, to which the authors have made a serious contribution. I recommend publishing an article in the journal "Applied Sciences".
Reviewer 2 Report
I find the paper “A study of an algorithm for the surface temperature forecast: from road ice risk to farmland application” interesting as an example of the application of a climate study for a practical purpose. In this study, the modelling and predicting of ice formation on the (paved and agricultural) surfaces is conducted. Since the geography of the research area has an impact on the research methodology I suggest to the authors to add the research area to the title.
I find that the introduction provides sufficient background, research design appropriate (although it is limited to the fact that the study is based on algorithms construction and testing), the methods adequately described and results are clearly presented. But several details could improve paper quality.
I think that Table 5 is unnecessary. All of the mentioned methods are commonly used and well-known; only the Nash-Sutcliffe efficiency should be presented by the formula and others could only be mentioned in the text as they already are.
The part in which the cost of the salt used to lower the freezing point of water is calculated (lines 467-486) is interesting but does not help to better understand or further analyse the algorithm for the ice-risk forecast. I suggest to shorten this part and to move it into the introductory part of the paper if the authors want to keep it in the manuscript.
There are the particular meteorological conditions for which the algorithms for LST and ice-risk forecast is tested (sky conditions, air temperature) so I suggest that all of them, as factors that impact the LST and conditions for the ice forming, be mentioned in the methods. Since the data for the cultivated fields were available for the growing season it should be stated the average duration of that period (in a particular part of Italy).
In the results, it is stated: “Intermediate conditions i.e. partly cloudy days, are here omitted because an accurate forecast of the incoming solar radiation is a hard task.” The authors should at least give a short insight in way how that problem could be solved in practical usage of the algorithm or what improvements are needed. Otherwise, the usage of the algorithm as presented is very limited. Or, for that part, the fact that the presence and risk of ice generally (in northern Italy) are high during clear nights only, should be pointed out.
In the lines 195-196, it is stated that Land Surface Temperature (LST) afterwards would be also called T-skin, but later in the manuscript, it is referred to as T-skin, skin temperature or surface temperature. Since there are several temperatures in the manuscript (air temperature, soil temperature, land surface temperature, output surface temperature) be sure that you use a correct name for the certain “type” of temperature.
I suggest not to use “following” or “above” when referring to figures and tables, but refer to the figure or table number, eg Line 388: instead “Data summarised in the above tables show…” there should be “Data summarised in the tables 6-9 show…”. There are several similar cases in the manuscript.
Also, there are several minor comments:
Lines 41-50: This part of the manuscript has different font type and alignment then the rest.
Line 88: There should be: … a study of an algorithm developed by the…
Figure 1: If it is possible, I would suggest to use a better map for Figure 1.
Line 340/495: Doesn’t the period from 1st of April 2017 to 31st of December 2018 lasts for 21 months?
Line 452: Should there be “growing season” instead of “summer season”?
Table 13: In the second row there should be: Cloudy days.
